# Synthesis and Characterization of Biodegradable Poly(vinyl alcohol)-Chitosan/Cellulose Hydrogel Beads for Efficient Removal of Pb(II), Cd(II), Zn(II), and Co(II) from Water

**DOI:** 10.3390/gels9040328

**Published:** 2023-04-13

**Authors:** Mona A. Aziz Aljar, Suad Rashdan, Abdulla Almutawah, Ahmed Abd El-Fattah

**Affiliations:** 1Department of Chemistry, College of Science, University of Bahrain, Sakhir P.O. Box 32038, Bahrain; 2Department of Materials Science, Institute of Graduate Studies and Research, Alexandria University, Alexandria 21526, Egypt

**Keywords:** chitosan, poly(vinyl alcohol), cellulose, composite hydrogels, heavy metals, adsorption, wastewater treatment

## Abstract

Globally, water contamination by heavy metals is a serious problem that affects the environment and human health. Adsorption is the most efficient way of water treatment for eliminating heavy metals. Various hydrogels have been prepared and used as adsorbents to remove heavy metals. By taking advantage of poly(vinyl alcohol) (PVA), chitosan (CS), cellulose (CE), and the process for physical crosslinking, we propose a simple method to prepare a PVA-CS/CE composite hydrogel adsorbent for the removal of Pb(II), Cd(II), Zn(II) and Co(II) from water. Structural analyses of the adsorbent were examined by Fourier transform infrared (FTIR) spectroscopy, scanning electron microscopy-energy dispersive X-ray (SEM-EDX) analysis, and X-ray diffraction (XRD). PVA-CS/CE hydrogel beads had a good spherical shape together with a robust structure and suitable functional groups for the adsorption of heavy metals. The effects of adsorption parameters such as pH, contact time, adsorbent dose, initial concentration of metal ions, and temperature on the adsorption capacity of PVA-CS/CE adsorbent were studied. The adsorption characteristics of PVA-CS/CE for heavy metals may be completely explained by pseudo-second-order adsorption and the Langmuir adsorption model. The removal efficiency of PVA-CS/CE adsorbent for Pb(II), Cd(II), Zn(II), and Co(II) was 99, 95, 92, and 84%, respectively, within 60 min. The heavy metal’s hydrated ionic radius may be crucial in determining the adsorption preference. After five consecutive adsorption–desorption cycles, the removal efficiency remained over 80%. As a result, the outstanding adsorption-desorption properties of PVA-CS/CE can potentially be extended to industrial wastewater for heavy metal ion removal.

## 1. Introduction

Water pollution has become a global environmental problem that threatens ecosystems and human health. Indiscriminate dumping of industrial wastes contaminates surface and groundwater [1,2,3,4]. Heavy metals and organic dyes are the most common contaminants found in industrial wastewater, which has a disastrous impact on sustainable ecosystems [5,6]. Heavy metals are characterized as metallic elements with a density that is higher than the density of water [5]. Heavy metals are non-biodegradable in nature as well; they can be absorbed and accumulated in sediments accordingly, and their level of toxicity and negative impact will increase on living organisms [7].

Due to their high carcinogenicity and toxic effect, heavy metals cause several diseases and significant physiological disorders such as anemia, hypophrenia, acute gastroenteritis, and nervous system damage to human health. Particularly, the presence of Pb(II), Ni(II), Cd(II), Cu(II), Co(II), and Cr(VI) in the effluent is a major environmental concern [8]. The harmful nature of heavy metal ions to public health and the natural ecosystem highlights the necessity of effective treatment of industrial wastewater [9].

Various treatment approaches have been developed for the removal of heavy metals ion exchange [10], adsorption [11,12], chemical precipitation [13,14], membrane separation [15,16], flocculation-coagulation [17], flotation [18], and electrochemical methods [19]. Amongst all the treatments proposed, adsorption using sorbents is one of the most popular methods since the proper design of the adsorption process will produce high-quality treated effluents. Adsorption is now recognized as an effective, efficient, and economic method for water remediation applications [20,21]. Furthermore, adsorption is a reversible process so that adsorbents can be sustained and reused through desorption [22]. Physical conditions such as pH, initial adsorbate and adsorbent dosage, contact time, and temperature are the most fundamental factors that affect the adsorption capacity of adsorbent materials. The optimization of these factors is very critical and should be considered first to design the adsorption process at a large scale [11,12,23].

For the removal of heavy metal ions, a variety of adsorbent materials have been used, including clay minerals, carbon nanomaterials, metal oxide nanoparticles, and polymeric hydrogels [23,24]. In general, the advantages of bio-based polymeric hydrogels as an adsorbent material have drawn considerable attention. They have created crosslinked three-dimensional network topologies, for instance, with excellent water sorption and retention capacities. Accordingly, they can capture and store different heavy metal ions found in water within their crosslinked network structure [20,23,24]. In addition, the functional groups in their network structures, such as carboxyl (−COOH), hydroxyl (−OH), and amine (−NH_2_), can afford the essential coordinating sites for the heavy metal ions. Additionally, significant advantages include easy fabrication, good mechanical properties, regeneration, and biodegradation. The term “biodegradability” describes a material’s capacity to degrade through the action of microorganisms into end products that are safe and non-toxic. The breakdown of hydrogels is affected by a number of factors, including molecular weight, hydrophilicity, and how the polymer interacts with water. The breakdown of polymers by solubilization and hydrolysis is also influenced by other environmental factors such as pH, salinity, and temperature. Enzymatic hydrolysis is another method by which hydrogels can be broken down, and this class of hydrogels includes biopolymers such as proteins and polysaccharides [23].

Hydrogels prepared from biopolymers such as polysaccharides (e.g., alginate, hyaluronic acid, cellulose, chitin, and chitosan) have widely been used in industrial applications because they are sustainable, biodegradable, economical, and eco-friendly [21,23,24]. Chitosan (CS) is a natural polymer (cationic polysaccharide). It is a high-abundance and cost-effective biopolymer that is biodegradable and biocompatible [25]. CS is used as an adsorbent for the removal of heavy metals because of the reactive functional groups (e.g., −OH and −NH_2_) in its molecular structure. These functional groups contribute to hydrophilicity and active adsorption sites through various types of interactions, including electrostatic, coordination, and hydrogen bonding [26,27]. However, similar to other polysaccharides, CS has some disadvantages, such as instability of its main structure, lack of mechanical strength, high solubility in acidic solutions, and deformation after drying [23,28]. Thus, modification of CS through blending with other materials by chemical or physical crosslinking is an essential step in improving mechanical resistance, reducing hydrophilicity, and stabilizing CS in extremely acidic environments [26]. In wastewater treatment, studies have shown that poly(vinyl alcohol) (PVA) can strengthen and support the structure of CS-polymers by forming hydrogen bonds with the NH_2_ groups of CS [26,29]. Moreover, PVA is used extensively to remove heavy metal ions from water due to its excellent chemical stability, biodegradability, biocompatibility, hydrophilicity, and high adsorption capacity [30].

Recently, a range of reinforcing agents and/or fillers, including inorganic, organic, or polymeric micro/nanomaterials, have been incorporated within polymeric hydrogels to improve and impart tailored properties to crosslinked networks. These fillers strengthen the polymer matrix leading to composite hydrogels offering enhanced absorption and mechanical properties. Therefore, finding completely renewable fillers having good compatibility with both PVA and CS so as to improve the properties of their blend still remains a challenge. Microcrystalline cellulose (CE) has received considerable attention as a reinforcing agent because of its favorable properties, such as large surface area, high mechanical strength, low density, non-toxic nature, biocompatibility, and biodegradability [31]. Thus, incorporating CE into the PVA-CS blended system offers efficient adsorbent features with high adsorption capabilities. However, to the best of our knowledge, little effort has previously been reported on hydrogel prepared by incorporating CE into PVA-CS via a physically crosslinking approach for heavy metal removal from water.

In view of these facts mentioned above, we show a simple, economic, and eco-friendly preparation of PVA-CS/CE composite hydrogels with good comprehensive performance and enhanced properties. The effect of CE content on the physical properties of the hydrogel, including the crystallinity and swelling ratio, was investigated. In addition, the PVA-CS/CE composite hydrogels were applied to adsorb different heavy metal ions, including Pb(II), Cd(II), Zn(II), and Co(II). The effects of pH, contact time, initial ion concentration, adsorbent dose, and temperature on the adsorption process were examined. Isotherms and kinetic models were studied to address the controlling mechanisms of heavy metal ions adsorption onto PVA-CS/CE. The prepared hydrogel can be utilized to collect and recover heavy metals from industrial wastewater and other hazardous conditions in a facile and effective manner.

## 2. Results and Discussion

### 2.1. Design Rationale of Composite Hydrogel Beads

The physically crosslinked PVA-CS/CE composite hydrogel beads were successfully prepared. Figure 1a–f shows the digital images of wet and dry hydrogel beads before and after the adsorption of Pb(II), Cd(II), Co (II), and Zn(II) respectively. Adsorption of metal ions onto hydrogel beads was confirmed by light-yellow and pink colors (Figure 1c–f). Wet beads were ~4.0 mm in diameter and had a consistently spherical structure with a smooth surface (Figure 1a). The beads’ shape was altered after air-drying, creating a rough surface with superior mechanical strength morphology. According to the research of Li and colleagues [32], this deformation is unavoidable as water evaporates from the wet hydrogel beads during the drying process, causing the hydrogel beads’ volume to shrink.

The percentage of CE fillers was fixed at 0.5%, and the ratio between the PVA and CS was 1:1. These parameters were chosen because they provide the best adsorption conditions for the removal of heavy metals and the stability of composite beads [32]. To evaluate the adsorption ability of prepared composite hydrogel beads, preliminary adsorption experiments were conducted. Figure 2 reveals a comparison of the percent removal of heavy-metal ions using two types of adsorbent beads: PVA-CS as the control hydrogel beads and PVA-CS/CE as the modified CS-based polymer hydrogel. Remarkably, the control beads had lower removal efficiency than the modified beads with CE. These results highlighted a significant correlation between CE incorporation and the efficiency of the removal of heavy metals. The use of CE as a filler in the structure of CS biopolymer can improve the adsorption capacity and mechanical stability of the adsorbent through surface accessibility of groups as a result of “pillaring” effects [33]. Moreover, the superior mechanical properties of CE relative to CS suggest that crosslinking of these two biopolymers may lead to improved and modified properties in their composite structure, in turn producing denser and mechanically stronger hydrogel beads [33,34]. Considering the enhancement we documented, PVA-CS/CE (0.5 wt.%) was chosen for further studies on adsorption.

### 2.2. Structure and Chemical Composition

The surface functional groups of PVA, CS, CE, and PVA-CS/CE composite hydrogel were detected by FTIR spectroscopy (Figure 3a). The broad band at 3649–3170 cm^−1^ was attributed to the stretching of hydrogen–bonded –OH and –NH groups of the polysaccharide unit for CS and PVA [1,29]. The peak at 2938 cm^−1^ was attributed to the vibrational stretching of –CH_2_. The peak at 1652 cm^−1^ corresponded to the stretching of C=N due to the reaction of the NH_2_ of CS and OH of PVA. The peak at 1590 cm^−1^ was associated with the stretching vibration of NH_2_ of CS [29,35]. The peak at 1458 cm^−1^ was ascribed to the CH_2_ vibration of crosslinked PVA. FTIR spectroscopy showed that CE had been introduced into the PVA-CS-based polymer. Figure 3b presents XRD patterns of PVA, CS, CE, and PVA-CS, as well as PVA-CS/CE composite hydrogel. The pure PVA exhibited the three typical diffraction peaks at 2θ = 19.5°, 22.9°, and 40.8°, corresponding to the (101), (200), and (102) planes of PVA crystallites, while CS presented a broad peak at 2θ around 21° because of its amorphous structure [35]. The XRD pattern of CE exhibits two distinct diffraction peaks, at 2θ = 16.4° and 22.8° due to the reflection from the different crystal planes of CE, such as (010) and (002) [35,36]. The PVA-CS and PV-CS/CE hydrogels show diffraction peaks at 2θ values of 25° to 40°, which may relate to greater crystallinity of hydrogels due to strong hydrogen bonding between the donor-acceptor groups (–NH_2_ and –OH) of CS and/or PVA or CE [34]. It was observed that the characteristic peaks of PVA are slightly shifted toward low 2θ values with the incorporation of CS and CE in the composite formulation. Moreover, all peaks of CE disappeared in PVA-CS/CE composite hydrogel because the content of CE was too low. From this evidence, the authors concluded that intermolecular interactions between PVA and CS are formed, which decreases the crystallinity of the PVA-CS/CE composite hydrogel [36].

### 2.3. Morphology

The morphologies and main element composition of PVA-CS/CE beads before and after adsorbing heavy metal ions were characterized through SEM and EDX, as illustrated in Figure 4. In all cases, SEM micrographs revealed spherical or slightly oval-shaped beads with robust, compact surfaces and well-crosslinked network structures. It can be observed that the CE is homogeneously distributed in the PVA-CS matrix without the presence of macroscopic aggregates. Before the adsorption of heavy metals, the SEM image illustrated the surface texture of PVA-CS/CE composite beads, having holes and small openings on the surface. This fact, along with the roughness of the pore walls, provides more contact sites and allows water to act as a transport channel for heavy metal ions from water media into the interior of the adsorbent network via the abundant pores structure [32]. In general, after adsorption, the surface morphology was observed to have much asperity and to be more coarsely grained. It is clearly noted that the surface structure remains crosslinked network structure. However, the channels became compact, and a reduction in hydrogel pores was significant. EDX provided an elemental composition of PVA-CS/CE beads before and after the uptake of metal ions. The EDX spectrum for the intact PVA-CS/CE hydrogel before immersion in the metal ions solutions did not show the characteristic peaks of heavy metal ions. However, the spectrum exhibited the emergence of carbon, oxygen, nitrogen, and sodium [37,38]. After being loaded with heavy metal ions, the EDX spectra clearly showed the presence of Pb(II), Cd(II), Zn(II), and Co(II) in PVA-CS/CE beads, which indicates the PVA-CS/CE beads being coated with the heavy metal ions successfully.

### 2.4. Swelling Behavior

The swelling behavior of polymeric hydrogels is a complicated phenomenon that includes three consecutive stages: diffusion of the solvent into the crosslinked network structure, chain relaxation inside the hydrated gels, and network expansion. The SR of hydrogels is influenced by hydrophilic groups, the characteristics of the solution, and the elasticity of the polymer network. Functional groups cause electrostatic repulsion inside the network, accordingly expanding the gel and ultimately resulting in equilibrium [39]. 

Figure 5 depicts the swelling behavior of the PVA-CS/CE composite hydrogel soaked in buffer solutions with different pH values of 3, 7, and 9 as a function of time. At pH 7 and 9, the PVA-CS/CE hydrogel had a lower SR than those at pH 3.

The NH_2_ groups of CS and OH groups of PVA can be protonated as (NH_3_^+^ and OH_2_^+^) in acidic solutions (pH 3), and the electrostatic repulsions induced by these ionic groups can increase the hydrophilicity of the hydrogel, thus expanding the hydrogel crosslinked networks. In contrast, swelling decreases under neutral (pH 7) and basic (pH 9) conditions due to the deprotonation of these ionic groups, thereby reducing repulsive forces and causing the formation of hydrogen bonds between the NH_2_ groups of CS and OH groups of PVA, which tends to increase the entanglement. As a result, the intramolecular interaction increases, and the pore size is reduced, which results in a reduction in the SR [28]. These results are in agreement with previously reported data [40].

### 2.5. Adsorption Studies of Heavy Metals

#### 2.5.1. Effect of pH on Metal Adsorption

The pH plays a significant role in controlling sorption because the proton concentration impacts metal solubility, the ionization of functional groups on hydrogel beads, and the contact with the adsorbent [41,42]. Based on the pH, metal ions can be present as cationic species or metal hydroxides [43]. Heavy-metal ions can be present in different oxidation states, and most heavy-metal ions form hydroxide precipitates in basic solutions [6,44]. In an acidic medium, hydrogen ions (H^+^) can protonate electron-rich groups such as NH_2_ to NH_3_^+^, carboxyl (COO^−^) groups, or OH groups. As a result of such protonation, the positive charge on the adsorbent surface increases. Nevertheless, in basic media, the presence of a negatively charged hydroxyl group (OH^−^) creates a negative surface on the adsorbent [6]. In addition, the overall charge on the surface of hydrogel beads is affected by the availability of functional groups on the surface and their reactivity towards H^+^ and OH^−^. Specifically, a net surface charge is shifted from net positive at low pH to net negative at high pH. So, the pH at which an adsorbent has zero charge (pH_PZC_) is an important parameter as it effectively predicts the surface charge at different pH values. The latter is not the neutral point (pH = 7); instead, it is the point at which the number of positive and negative ions are equal. The pH_pzc_ is controlled by the type and number of functional groups present on the hydrogel surface. A net positive charge is obtained below pH_pzc_, which indicates that the adsorbent can adsorb the negatively charged ions via electrostatic forces. In contrast, above pH_pzc_, the net surface charge of the adsorbent is negative, which suggests that positive ions are attracted by electric forces [45]. 

Figure 6 shows the surface-charge density of PVA-CS/CE composite beads as a function of pH. The intersection of the curve with the *x*-axis at ΔpH (pH_initial_ − pH_final_) equaling zero gives pH_pzc_. The pH_pzc_ of the adsorbent was measured at pH 5. If pH > 5, then the surface of the adsorbent is negatively charged, and if pH < 5, then the surface could have a positive charge. The pH_pzc_ is an important factor in adsorption because it explains the behavior of heavy metals on the surface of the adsorbent.

The adsorption of Pb(II), Cd(II), Zn(II), and Co(II) ions on PVA-CS/CE composite hydrogel was measured at pH 3–9 (Figure 7a). Removal of Pb(II) and Cd(II) was increased upon increasing the initial pH of the solution from 3 to 6, and maximum removal of Pb(II) and Cd(II) was observed at pH 6. Conversely, the maximum removal of Zn(II) and Co(II) on the PVA-CS/CE composite hydrogel was achieved at pH 8. The pH_pzc_ of PVA-CS/CE hydrogel was 5, so the removal efficiency of Pb(II), Cd(II), Zn(II), and Co(II) increased if pH > 5. Moreover, the low adsorption of metals in acidic media (pH < 7) is attributed to the competition of H^+^ with the cationic forms of metal ions onto adsorption sites. Importantly, at low pH, an electrostatic repulsion is formed due to the adsorption of cationic metal ions and the positive surface of the composite, which reduces the removal efficiency of the metal [43,46]. However, in basic media, a small number of metal ions are present in their cationic form, whereas most metal ions change to become metal hydroxides and precipitate. In this sense, the adsorption of ions increases in basic media. The effect of pH in our study is similar to that reported previously [43,46]. 

#### 2.5.2. Effect of Contact Time

Contact time is an important parameter in batch adsorption [47]. The effects of contact time on the removal of Pb(II), Cd(II), Zn(II), and Co(II) by the PVA-CS/CE composite hydrogel were studied between 20 min and 200 min (Figure 7b). A three-step sorption mechanism was observed. The first step was a rate-limiting step and corresponded to surface diffusion. The second step was also a rate-limiting step and was related to the sorption of metal ions and adsorbent beads. The final step was the equilibrium [48]. The experimental findings demonstrate that PVA-CS/CE adsorbs heavy metal ions in the medium at a very rapid rate. Pb(II), Cd(II), and Zn(II) all reached adsorption equilibrium states in less than an hour, whereas Co(II) took 140 min. For Pb(II), Cd(II), Zn(II), and Co(II), the removal efficiency of PVA-CS/CE was approximately 99, 95, 92, and 76%, respectively, at equilibrium. It is worth mentioning that the rapid removal of metallic ions on the adsorbent surface at the initial stage of the process can be attributed to the availability of more active adsorption sites at the beginning of adsorption, which permits the rapid accumulation of metallic ions on the adsorption sites. As this process continues, the active sites are filled, the accessibility of metal ions to the active sites becomes low due to the saturation of active sites, and eventually, adsorption approaches an equilibrium at a certain time [24,36].

#### 2.5.3. Effect of the Initial Concentration of Heavy Metals

With regard to the removal of metal ions from industrial wastewater, the initial concentration of metallic ions is an important factor in observing the adsorption capacity of adsorbent composite beads [24]. Adsorption tests were also carried out with other parameters fixed at initial heavy metal concentrations of 50 to 200 mg/L (Figure 7c). The removal efficiency of metallic ions was a function of their initial concentration. It could be noticed that the removal efficiency decreased with increasing initial heavy metals concentration. For example, at initial Zn(II) concentrations of 50, 150, and 200 mg/L, the removal rate of beads was approximately 92, 85, and 80%, respectively. This meant that even at low metal ion concentrations, hydrogel beads had an abundance of active adsorption sites, resulting in high adsorption for the majority of the metal ions. However, as the concentration of metal ions increased, adsorption sites would become saturated [37,49]. 

#### 2.5.4. Effect of Adsorbent Dose

The adsorbent dose is an important factor when determining the maximum number of ions adsorbed with respect to functional groups [44]. The effect of the adsorbent dose on the removal efficiency of heavy metal ions was investigated in the range from 0.1 to 0.5 g, and the results are presented graphically in Figure 7d. It was found that for a constant metal ion concentration, an increase in percent removal was observed as the adsorbent dose increased until saturation was reached. For example, in the case of Pb(II) and Cd(II), the percent removal increased from 84 to 98% and 79 to 93%, respectively, as the adsorbent dose increased from 0.1 g to 0.5 g. The reason for this phenomenon may be attributed to an increase in the number of active adsorption sites when the adsorbent mass increased [44,50]. Similar results have been reported by Esrafili et al. [43]. 

Moreover, the experimental results (Figure 7c) showed that when the initial concentration of the metal ion was 50 mg/L, and the PVA-CS/EC amount was 0.2 g, the removal efficiency of Pb(II), Cd(II), Zn(II), and Co(II) was 93, 88, 83, and 57%, respectively. The affinity of adsorption by PVA-CS/EC was discovered to decrease in the following order: Pb(II) > Cd(II) > Zn(II) > Co(II). The hydrated metal ion radius effect is responsible for these observations [50,51]. Metal ions in an aqueous medium form are hydrated with water molecules, and the ionic radii of Pb(II), Cd(II), Zn(II), and Co(II) are 1.19, 0.96, 0.74, and 0.74 Å, respectively. Due to steric overcrowding on the surface, when the ionic radius of a metal ion is large, it is simple to react with water molecules and produces a fast saturation of adsorption sites [50]. Due to the hydrophilicity of PVA-CS/CE, metal ions from the surrounding environment are readily attracted to the adsorbent in the form of hydrated ions, leading to favorable interactions between cationic pollutants and the anionic adsorbent surface [52]. Given that Pb(II) has the biggest ionic radius and has achieved the maximum adsorption rate among the four heavy metal ions. It seems that the difference in removal efficiency order and adsorption capacity may be related not only to metal ions’ properties but also to physical-chemical properties of the adsorbent, such as morphology, surface area, and pores distribution as well as hard-hard and soft-soft preferences between functional groups and metal ions [50]. Therefore, a material that is a good adsorbent for one adsorbate may not be the right adsorbent for another.

#### 2.5.5. Effect of Temperature

The effect of temperature on the removal of heavy-metal ions by PVA-CS/CE from an aqueous solution was investigated (Figure 7e). The adsorption efficiency increased with an increase in the temperature of the solution from 20 °C to 30 °C for all the metal ions because of an increase in the rate of diffusion and mobility of ions toward adsorption sites [51,52]. However, the percent removal of Pb(II), Cd(II), Zn(II), and Co(II) decreased as the temperature increased to 40 °C. Metal ions adsorbed on the surface of a hydrogel can be expelled into an aqueous solution as the saturation point of a heated solution decreases due to ion desorption from the gel surface [53]. The reduction in the number of ions removed with increasing temperature showed that adsorption was exothermic and favored a low temperature. Our results are in complete agreement with those of other scholars [51,53,54]. 

#### 2.5.6. Reusability of the Adsorbent

In addition to an adsorbent’s exceptional ability to bind to heavy metal ions, regeneration, and reusability are important traits of a top-notch adsorbent. If the adsorbent cannot be used repeatedly or if its adsorption efficiency is dramatically decreased after repeated use, its application value is diminished. In this regard, the adsorption ability of composites was tested for five successive cycles of sorption-desorption. First, the adsorbent beads were washed with HCl for 15 min. Thereafter, the adsorbent beads were filtered and dried. The collected adsorbent was utilized for the removal of Pb(II), Cd(II), Zn(II), and Co(II) for five consecutive experiments, and the recovery experiment was repeated for each cycle of adsorption (Figure 7f). After the fifth adsorption process, the adsorbent could remove up to 80% of heavy-metal ions. This phenomenon might have occurred because, under an acidic condition, repulsion between protonated NH_2_ groups and metal ions takes place, which accelerates desorption [55]. Thus, the as-prepared hydrogel beads were reusable for five cycles of adsorption and could be considered adsorbents for the adsorption of heavy metal ions.

### 2.6. Adsorption Isotherms

Sorption data were calculated using isotherm equations to describe the adsorption behaviors of the investigated heavy metal ions in an aqueous solution. The equations consider the adsorbent’s surface property and affinity, from which the theoretical Langmuir and Freundlich models were developed to calculate the adsorption isotherms. The Freundlich model depicts the adsorption behavior on heterogeneous adsorbent surfaces, assuming the formation of multilayer adsorption on the surface. The Langmuir isotherm model defines the adsorption behavior associated with monolayer adsorption on the surface of an adsorbent [56]. Table 1 shows the fitting parameters for the Freundlich and Langmuir isotherms for the adsorption of Pb(II), Cd(II), Zn(II), and Co(II) onto PVA-CS/CE adsorbent beads. Figure 8a,b show the linear relationships between q_e_ and C_e_ of the Langmuir and Freundlich correlations of each metal ion.

Adsorption of Pb(II), Cd(II), Zn(II), and Co(II) by the PVA-CS/CE hydrogel demonstrated a superior correlation coefficient (R^2^) defined by the Langmuir equation when compared to the regression values of the isotherm. These results suggested that a monolayer of each adsorbate formed on the surface of the adsorbents, supporting the Langmuir isotherm as a more accurate description of the adsorption behaviors of Pb(II), Cd(II), Zn(II), and Co(II) onto the PVA-CS/CE hydrogel [57]. When K_F_ and 1/n, two parameters related to the Freundlich isotherm, were examined, a high K_F_ value suggested a good capacity for the adsorption of the metal ions. Moreover, the 1/n value in the 0.1–1.0 range suggested a successful adsorption process. The quantified q_m_ values obtained after the Langmuir isotherm were consistent with the trend identified from these experimental data on adsorption optimization [58].

### 2.7. Adsorption Kinetics

The adsorption of Pb(II), Cd(II), Zn(II), and Co(II) onto PVA-CS/CE stimulated the formation of metal-adsorbent complexes by coordinating these cationic ions with the free electrons of nitrogen and oxygen atoms [43]. The transfer of metal ions onto the surface of the adsorbents defines the rate of adsorption [59]. The adsorption kinetics of metallic ions was studied using pseudo-first-order and pseudo-second-order kinetic models. Figure 9a,b show the linear plots for pseudo-first-order and pseudo-second-order kinetic models, respectively, and Table 2 displays the time-dependent parameters of adsorption kinetics and the results of the fitting. The pseudo-second-order kinetic model best characterized the adsorption of Pb(II), Cd(II), Zn(II), and Co(II) onto PVA-CS/CE adsorbent, according to the interpretive results. For all heavy metals, the R^2^ values of the pseudo-second-order kinetic model for adsorption were over 0.99, and for Cd(II) and Zn(II) ions, they even equaled one. The R^2^ values for all metal ions in the pseudo-first-order kinetic model, however, were lower than 0.99. Moreover, Pb(II), Cd(II), Zn(II), and Co(II) ion equilibrium adsorption capacities (q_e_) predicted from the pseudo-second-order were 4.9671, 4.9531, 4.0566, and 4.2694 mg/g, respectively, which were compatible with the experimental results (Table 2). However, for heavy metals, the estimated q_e_ values from the pseudo-first-order significantly diverged from the experimental results. Therefore, the pseudo-second-order kinetic model was more adapted for PVA-CS/CE adsorption of heavy metals than the pseudo-first-order kinetic model [55].

### 2.8. Adsorption Mechanism of Heavy Metal Ions

Based on the proposed structure of the PVA-CS/CE composite hydrogel that contained several hydroxyl and amino groups, electrostatic and coordination interactions were possible mechanisms for the adsorption of metal ions, including Pb(II), Cd(II), Zn(II), and Co(II) on the surface of the hydrogel [23,51]. At pH > pH_PZC_, functional groups, such as –OH, are deprotonated that creates anions (such as –O^–^) on the adsorbent surface, resulting in electrostatic–attractive interactions between cationic metal ions contaminants and the anionic adsorbent surface (Figure 10). Moreover, cations can attract atoms in functional groups with lone pair electrons (i.e., O and N) in outer orbitals, resulting in the adsorption of cations on the adsorbate surface via coordination interaction.

Table 3 compares the maximum adsorption capacity values of the PVA-CS/CE hydrogel beads synthesized in this study and other previously published data. For the metal ion removal application, the prepared PVA-CS/CE showed good adsorption characteristics. It is simple to fabricate and has excellent removal rates for Pb(II), Cd(II), Zn(II), and Co(II) ions as well as quick adsorption equilibrium, stability, and reusability.

## 3. Conclusions

An easy physical crosslinking route has been proposed to prepare the environmentally friendly PVA-CS/CE composite hydrogel beads with good adsorption performance for Pb(II), Cd(II), Zn(II), and Co(II) from aqueous solutions. The incorporation of CE into the PVA-CS matrix significantly improved the adsorption capacities of the metal ions. The experimental findings showed that the optimal pH for Pb(II) and Cd(II) was 6, and 8 for Zn(II) and Co(II), and that adsorption equilibrium was reached in 60 min for Pb(II), Cd(II), and Zn(II), but 140 min for Co(II). Pb(II) > Cd(II) > Zn(II) > Co(II) is the order of heavy metal affinity for PVA-CS/EC adsorbent. Adsorption isotherm data fit well with the Langmuir isotherm model, indicating a homogeneous adsorption mechanism with monolayer-deposited ions, while kinetic data of each metal ion preferably followed the pseudo-second-order kinetic model. The removal efficiencies for Pb(II), Cd(II), Zn(II), and Co(II) during the fifth reuse cycle were 94.41, 91.72, 89.03, and 80.62%, respectively. This demonstrates that the PVA-CS/CE adsorbent may be recycled about five times before being used again, which lowers waste generation and has a valuable use for the treatment of industrial wastewater. Even though hydrogels have received a lot of research, several areas still need to be explored. Currently, the only application for bio-based hydrogel adsorbents in heavy metal removal is on a lab scale. As a result, additional study is required to scale up for a large-scale application in order to understand the cost effectiveness and adsorption efficacy better. In addition, the research should focus on the creation of bio-based hydrogel materials with high mechanical strength that are simpler to separate from the liquid phase for water remediation.

## 4. Experimental

### 4.1. Materials

The materials used for this study were PVA (molecular weight (MW) = 13,000–23,000 gmol^−1^; 89 mol% hydrolyzed), CS (93% degree of deacetylation, Mw = 200,000 gmol^−1^), CE (microcrystalline powder with an average particle size of 45 ± 5 μm), acetic acid (CH_3_COOH), hydrochloric acid (HCl), nitric acid (HNO_3_), and sodium hydroxide (NaOH). All these materials were purchased from MilliporeSigma (Burlington, MA, USA) and used without further purification. Stock solutions (1000 ppm) of standard solutions of lead, cadmium, cobalt, and zinc were purchased from Romil (Cambridge, UK). A series of standard solutions of Pb(II), Cd(II), Zn(II), and Co(II) (1.0, 2.0, 3.0, 4.0, and 5.0 mg/L) were prepared from the stock solution and diluted by HNO_3_ (2.0 wt.%). All polymeric solutions were prepared in deionized water (0.055 μScm^−1^) except CS with CH_3_COOH (2.0 wt.%).

### 4.2. Preparation of PVA-CS/CE Hydrogel Beads

The PVA-CS/CE hydrogel beads were fabricated via the physical crosslinking method, as described in a previous study [20]. PVA (4 g) was dissolved in deionized water (50 mL), and CS (2 g) was dissolved in CH_3_COOH solution (2% *v/v*, 100 mL). PVA and CS solutions were mixed under continuous stirring at 70 °C for 2 h to produce a homogenous solution. CE powder (0.5% *w/w*) was added into the stirring homogeneous solution at 70 °C. The stirring was continued at 70 °C for another 3 h. After that, the solution was poured dropwise into NaOH solution (10%, 300 mL) to form hydrogel beads and maintained overnight. The hydrogel beads were collected by filtration, washed with deionized water, and dried at room temperature.

### 4.3. Characterization of PVA-CS/CE Composite Hydrogels

Fourier transform infrared (FTIR) spectroscopy was used to characterize the functional groups on the surface of adsorbents and to observe interactions in the adsorbent–adsorbate system. A Prestige-21 instrument (Shimadzu, Beijing, China) was used to record the IR spectra of the adsorbent material before and after adsorption. The adsorbent was finely ground with KBR and pressed to make pellets before FTIR spectroscopy. After that, the spectra were examined in transmission mode from 400 cm^−1^ to 4000 cm^−1^. A scanning electron microscope (SEM) was used to investigate the surface morphology of PVA-CS/CE composite hydrogel beads before and after the adsorption of metallic ions. SEM images were obtained using a JSM-5300 instrument (Jeol, Tokyo, Japan) working at 20 kV. Before SEM, samples were coated with gold to a thickness of 0.04 μm using sputter-coating equipment (JFC 1100 E; Jeol) after being ultrasonically washed for 30 s. The elemental compositions of PVA-CS (control hydrogel) and PVA-CS/CE composite hydrogel beads before and after adsorption of heavy metal ions were determined by energy dispersive X-ray (EDX) microanalysis attached to the SEM system. Uncoated samples were analyzed at 15 kV for 60 s. Crystal structural analysis of the pure PVA, CS, CE, and their composite hydrogel was performed by powder X-ray diffraction (XRD) measurements using a Diffractometer (XRD-7000, Shimadzu, Japan). The X-ray beam was Cu-Kα radiation (λ = 0.1542 nm), operated at 40 kV and 30 mA. The XRD pattern was recorded in the 2θ range from 10° to 100° with a scanning rate was 5° per min.

### 4.4. Swelling Ratio Test

The swelling ratio (SR) provides information about the water absorption capacity of a hydrogel. SR of PVA-CS/CE composite hydrogel beads was measured in deionized water and solutions with different pH values (3, 7, and 9). Approximately 0.5 g of dried hydrogel beads were immersed in deionized water (100 mL), and the pH was adjusted to 3, 7, or 9 at room temperature until the beads swelled to equilibrium. Over specific time periods, the swollen beads were removed and wiped with tissue paper to remove residues of surface water. Then, the mass of the fully swollen samples was determined. Equation (1) was used to calculate the SR in various solutions [29].
(1)$$R\;(\%)=\left[\frac{W_{w}-W_{d}}{W_{d}}\right]\times 100$$
where W_w_ and W_d_ are the weight of wet hydrogel and dry hydrogel, respectively.

### 4.5. Adsorption of Heavy Metals

To investigate the removal percentage and adsorption capacities of the PVA-CS/CE adsorbent for heavy metal ions, 0.5 g of adsorbent was placed in a 100-mL Erlenmeyer flask containing 50 mL aqueous medium with 50 mg/L initial concentration of Pd(II), Cd(II), Zn(II), and Co(II). Each batch adsorption was kept on a shaker with a speed of 170 rpm at 25 °C. Then, aliquots (5 mL) were collected after 200 min, passed through filter paper (0.45 μm), and analyzed using inductively coupled plasma-optical emission spectroscopy (ICP-OES, Perkin Elmer Avio 220, USA) to measure Pb(II) (at 220.35 nm), Cd(II) (228.8 nm), Zn(II) (206.2 nm), and Co(II) (228.61 nm). The removal percentage and adsorption capacity of the PVA-CS-/CE adsorbent for each metal ion was calculated from Equations (2) and (3), respectively [70].
(2)$$\mathrm{Removal}\;(\%)=[\left(C_{0}-C_{e}\right)/C_{0}]\;\times \;100\;$$
(3)$$q_{e}=\left(\left(C_{0}-C_{e}\right)\times \;V\right)/m$$
where q_e_ = equilibrium adsorption capacity of the synthesized adsorbent (mg/g), C_0_ = initial concentration of the heavy metal in the aqueous medium (mg/L), C_e_ = concentration of the heavy metal in the aqueous medium after adsorption (mg/L), V = volume of the heavy metal ion solution, and m = mass of the adsorbent (g).

The effect of several parameters, such as pH (3–9), initial metal ion concentrations (50–200 mg/L), adsorbent dose (0.1–0.5 g), contact time (25–200 min), and temperature (20–40 °C) on the adsorption was studied. Adsorption experiments with various initial concentrations ranging from 50 to 200 mg/L were carried out to investigate the optimized initial concentration of the metal ion. Heavy metal solutions’ pH was raised from 3 to 9. After adjusting the pH with 0.1 M HNO_3_ or 0.1 M NaOH solutions, the mixture was agitated continuously for 25–200 min. Isothermal experiments at 20, 30, and 40 °C were carried out to investigate the effect of temperature. The adsorbent dose ranged from 0.1 to 0.5 g in this set of experiments.

### 4.6. Adsorbent Reusability

Multiple sorption-desorption cycles were used to assess the suitability of recovering PVA-CS/CE hydrogel beads. Hydrogel beads (0.5 g) were added to a metallic ion solution (50 mg/L, 50 mL) at pH 6 (Pb and Cd) and pH 8 (Zn and Co). At 30 °C, the solutions were agitated in a water-bath shaker for 200 min. The saturated metal-loaded beads were collected, washed in deionized water, and dried at room temperature for 48 h. For desorption, the dried beads were placed in a 50 mL of 0.1 M HCl solution and agitated at 170 rpm for 30 min. The beads were filtered from the solution, rinsed several times with deionized water to remove surface metal, and dried at room temperature for 48 h before reuse. The same adsorption method was carried out using the recycled adsorbent beads under identical circumstances. There were five sorption-desorption cycle tests performed. Equation (4) was used to determine how effective adsorbent beads were in being reused [71]: (4)$$\mathrm{Reuse}\;\mathrm{efficiency}\;(\%)=\frac{q_{n}}{q_{1}}\times 100\%$$
where q_1_ and q_n_ (mg/g) represent the initial and fifth-time adsorption capacities of metal ions by reused PVA-CS/CE composite hydrogel, respectively.

### 4.7. Adsorption Isotherms

Adsorption isotherms reveal how well adsorbate ions coordinate with the adsorbent and when the equilibrium point is reached between the adsorbed metal ions and residual ions in a solution [72]. Two adsorption isotherms were used to investigate adsorption mechanisms, Langmuir and Freundlich, as described in Equations (5)–(7), respectively [72].
(5)$$\frac{C_{e}}{q_{e}}=\frac{1}{K_{L}q_{m}}+\frac{C_{e}}{q_{m}}$$
(6)$$R_{L}=\frac{1}{1+K_{L}C_{0}}$$
(7)$${\ln\;q}_{e}={\mathrm{lnK}}_{F}+\frac{1}{n}{\;\mathrm{lnC}}_{e}\;$$
where C_e_ (mg/L) is the concentration of metal ions at equilibrium, q_e_ (mg/g) is the number of metal ions adsorbed at equilibrium and saturation, q_m_ (mg/g) is the saturated adsorption capacity, K_L_ (L/mg) and K_F_ [(mg/g)(L/mg)^1/n^] represent the Langmuir adsorption equilibrium constant and Freundlich constant, respectively, and R_L_ refers to the degree of adsorption for favorable and multilayer adsorption; 1/n should be between 0 and 1.

### 4.8. Adsorption Kinetics

Kinetic models were used to study the adsorption mechanism of Pb(II), Cd(II), Zn(II), and Co(II) on PVA-CS/CE composite hydrogel beads. Experimental data were modeled by pseudo-first-order and pseudo-second-order kinetic models. The linear forms of both models are explained in Equations (8) and (9) [73]:(8)$$\ln\;\left(q_{e}-q_{t}\right)={\mathrm{lnq}}_{e}-K_{1}t$$
(9)$$\frac{t}{\mathrm{qt}}=\frac{1}{K_{2}q_{e}^{2}}+\frac{1}{q_{e}}\;t$$
where q_t_ and q_e_ (mg/g) are the amount of metal removed per unit mass of adsorbent (mg/g) at a certain time t and at equilibrium, respectively. K_1_ (1/min) is a pseudo-first-order rate constant, and K_2_ (g/mg·min) is the rate constant of a pseudo-second-order reaction

### 4.9. Statistical Analysis

A one-way ANOVA test was used to see whether there were any significant differences between data from the experimental groups. This was followed by Tukey’s posthoc test and a Student t-test. Each experiment was run in at least five repetitions, and all data were provided as mean and standard deviation (M ± SD). A *p*-value of 0.05 or less was regarded as significant.

## Figures and Tables

**Figure 1 gels-09-00328-f001:**
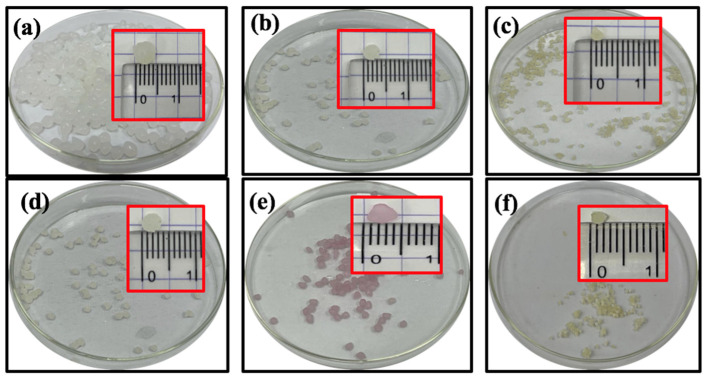
PVA-CS/CE composite hydrogel beads (**a**) before drying, (**b**) after 48 h of air drying, as well as dried beads after adsorption of (**c**) Pb(II), (**d**) Cd(II), (**e**) Co(II), or (**f**) Zn(II). Insets show the size of beads.

**Figure 2 gels-09-00328-f002:**
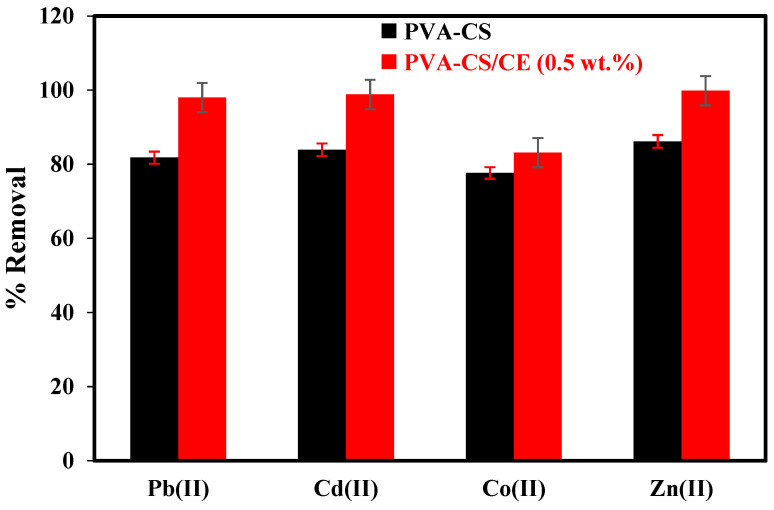
Effect of CE content (0, and 0.5 wt.%) on percent removal of Pb(II), Cd(II), Co(II), and Zn(II) at hydrogel beads (C_0_ [heavy metals] = 50 mg/L, volume = 50 mL, pH (Pb and Cd) = 6, pH (Co and Zn) = 8, bead dose = 0.5 g, time = 200 min, temperature = 30 °C).

**Figure 3 gels-09-00328-f003:**
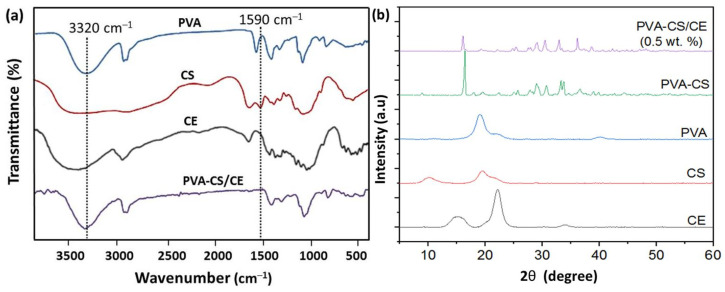
(**a**) FTIR spectra and (**b**) X-ray diffraction patterns of PVA, CS, CE, PVA-CS control hydrogel, and PVA-CS/CE composite hydrogel.

**Figure 4 gels-09-00328-f004:**
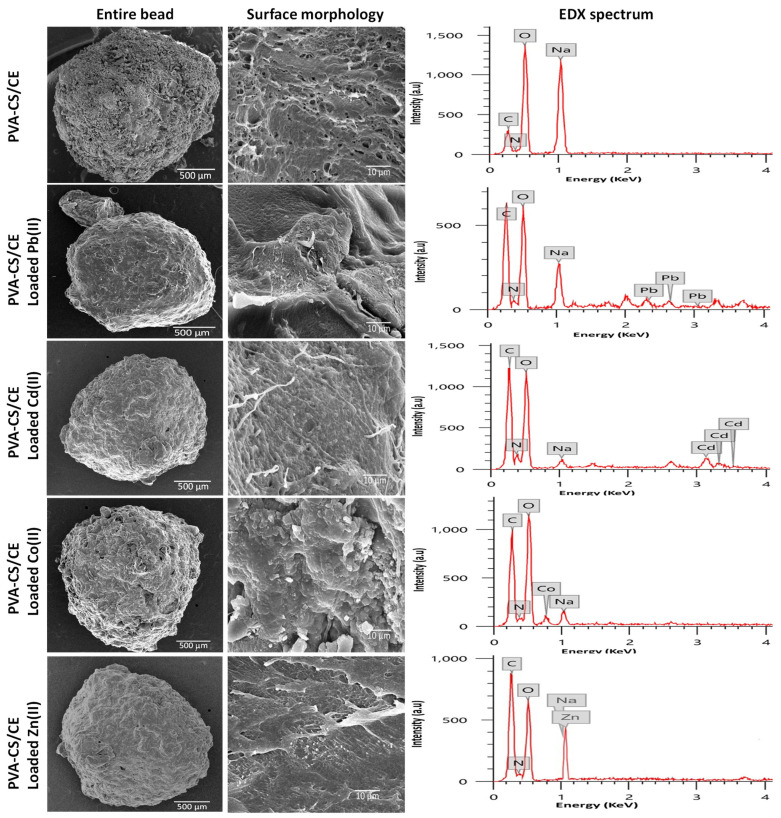
SEM images of the entire surface of PVA-CS/CE hydrogel beads before and after adsorption of heavy metal ions as well as EDX spectra of PVA-CS/CE hydrogel beads.

**Figure 5 gels-09-00328-f005:**
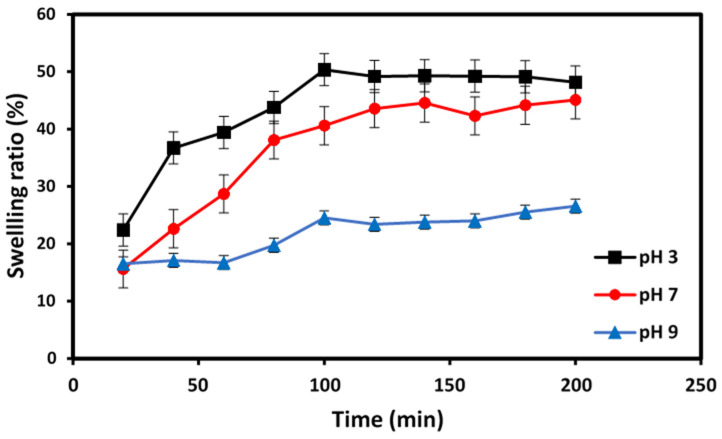
Swelling behavior of PVA-CS/CE composite hydrogel beads at different pH values.

**Figure 6 gels-09-00328-f006:**
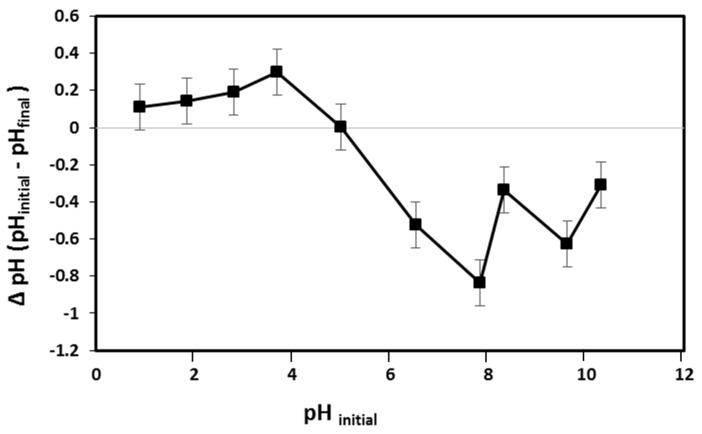
Determination of point zero charge (pH_pzc_) of PVA-CS/CE composite hydrogel beads.

**Figure 7 gels-09-00328-f007:**
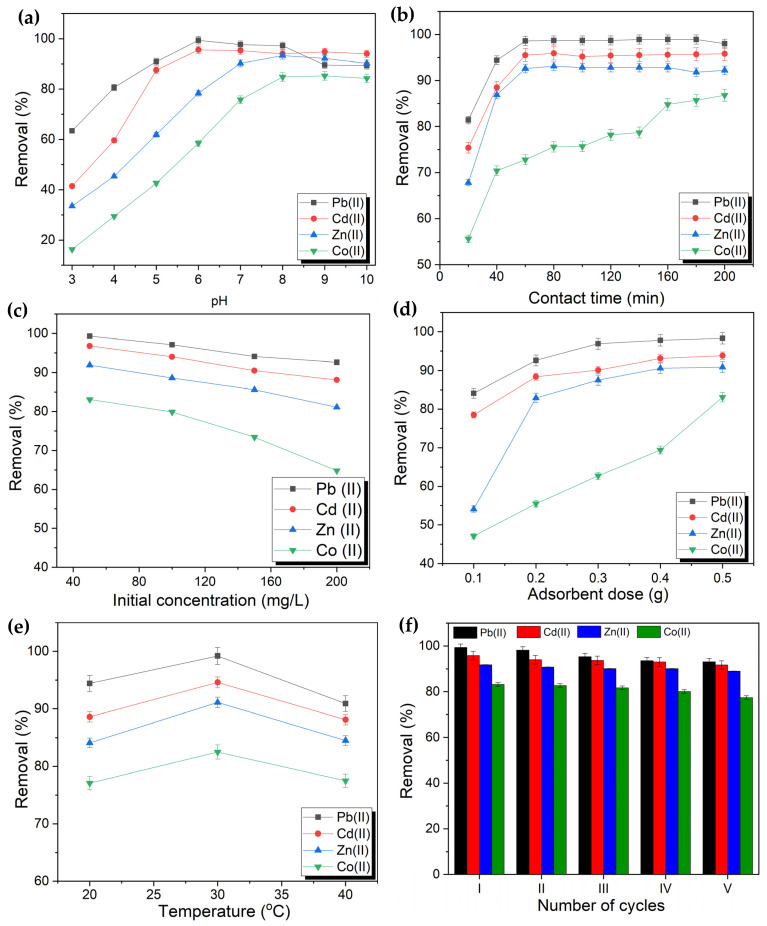
Effects of different factors on the removal efficiency of Pb(II), Cd(II), Zn(II), and Co(II) onto PVA-CS/CE composite hydrogel beads: (**a**) pH (C_o_[Metals] = 50 mg/L, V = 50 mL, bead dose = 0.5 g, pH = 3–10, t = 200 min, T = 30 °C), (**b**) Contact time (C_o_[Metals] = 50 mg/L, V = 50 mL, pH = 6 (Pb and Cd) and pH = 8 (Zn and Co), bead dose = 0.5 g, T = 30 °C), (**c**) Initial concentration of metals (V = 50 mL, pH = 6 (Pb and Cd) and pH = 8 (Zn and Co), bead dose = 0.5 g, t = 200 min, T = 30 °C), (**d**) Adsorbent dose (C_o_[Metals] = 50 mg/L, V = 50 mL, pH = 6 (Pb and Cd) and pH = 8 (Zn and Co), t = 200 min, T = 30 °C), (**e**) Temperature (C_o_[Metals] = 50 mg/L, V = 50 mL, pH = 6 (Pb and Cd) and pH = 8 (Zn and Co), bead dose = 0.5 g, t = 300 min.), and (**f**) Reusability (C_o_[Metals] = 50 mg/L, V = 50 mL, pH = 6 (Pb and Cd) and pH = 8 (Zn and Co), bead dose = 0.5 g, t = 200 min, T = 30 °C).

**Figure 8 gels-09-00328-f008:**
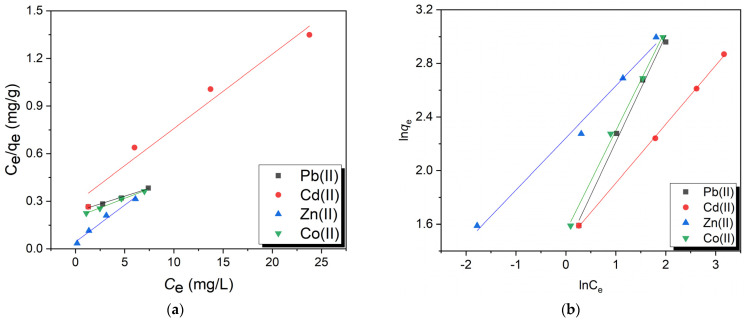
(**a**) Langmuir and (**b**) Freundlich adsorption isotherm models for the adsorption of Pb(II), Cd(II), Co(II), and Zn(II) onto PVA-CS/CE composite hydrogel beads.

**Figure 9 gels-09-00328-f009:**
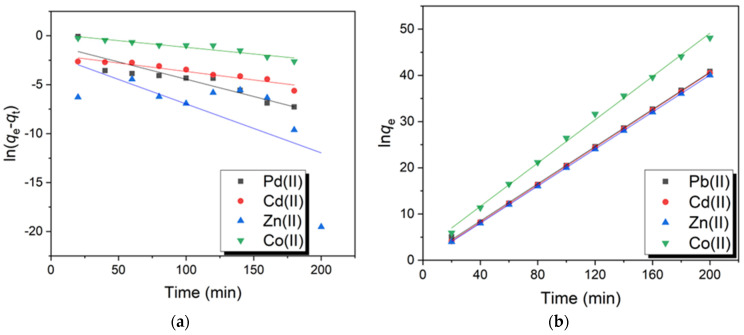
(**a**) Pseudo-first-order and (**b**) pseudo-second-order kinetic models for the adsorption of Pb(II), Cd(II), Zn(II), and Co(II) onto PVA-CS/CE composite hydrogel beads.

**Figure 10 gels-09-00328-f010:**
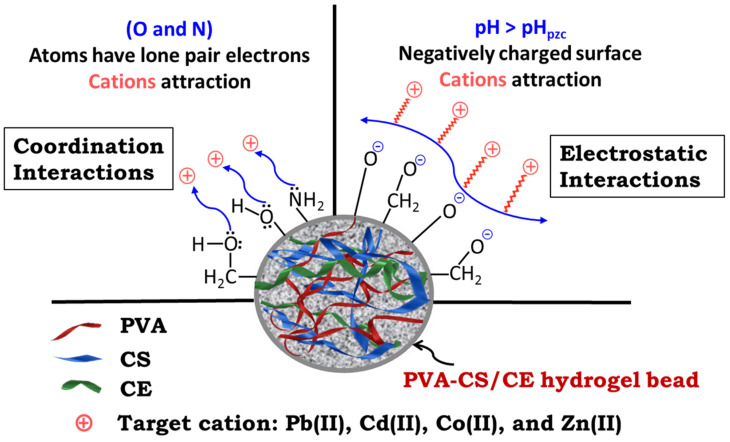
Adsorbent–adsorbate interaction mechanisms for the heavy metal ions decontamination of water by PVA-CS/CE composite hydrogel.

**Table 1 gels-09-00328-t001:** Adsorption isotherm parameters of Pb(II), Cd(II), Zn(II), and Co(II) onto PVA-CS/CE.

Model	Parameter	Pb(II)	Cd(II)	Zn(II)	Co(II)
Langmuir equation	q_m_ (mg/g)	50.54	41.82	26.74	20.53
	R_L_	0.054–0.187	0.029–0.109	0.015–0.014	0.043–0.141
	R^2^	0.993	0.968	0.996	0.994
	K_L_	0.087	0.163	1.422	0.121
Freundlich equation	K_F_ (L/mg)	5.26	4.34	9.22	4.67
	1/n	0.688	0.438	0.407	0.753
	R^2^	0.979	0.999	0.993	0.997

**Table 2 gels-09-00328-t002:** Kinetic parameters for the sorption of metal ions in a batch-adsorption system.

Model	Parameter	Pb(II)	Cd(II)	Zn(II)	Co(II)
Pseudo-first order	R^2^	0.8434	0.9098	0.5026	0.9031
	q_e_ (mg/g)	0.4045	0.1476	0.5277	0.8332
	K_1_ (min^−1^)	0.0353	0.0173	0.0584	0.0136
Pseudo-second order	R^2^	0.9996	1.00	1.00	0.9974
	q_e_ (mg/g)	4.9671	4.9531	4.0566	4.2694
	K_2_ (g/mg.min)	0.0953	0.2748	0.1625	0.0343
	q_e_ (experimental)	4.9321	4.9423	4.9901	4.1561

**Table 3 gels-09-00328-t003:** Comparison of adsorption capacities of Pb(II), Cd(II), Zn(II), and Co(II) for different Adsorbents.

Metal	Adsorbent	Adsorption Capacity (mg/g)	Reference
**Pb(II)**	Chitosan/polyvinyl alcohol/β-cyclodextrin membrane	13.44	[60]
	Chitosan/activated carbon/polyvinyl alcohol composite	22.47	[20]
	Chitosan/cellulose composite	26.81	[61]
	Ethylenediamine modified chitosan	30.57	[62]
	PVA-CS/CE composite hydrogel beads	50.54	This study
**Cd(II)**	Chitosan/polyvinyl alcohol/polyvinylpyrrolidone hydrogel	08.36	[63]
	Poly(maleic acid)-grafted crosslinked chitosan microspheres	39.21	[64]
	Chitosan-coated cotton fibers	15.74	[65]
	PVA-CS/CE composite hydrogel beads	41.82	This study
**Zn(II)**	Chitosan/cellulose composite	19.81	[61]
	Xanthate-modified magnetic chitosan composite	20.8	[66]
	Ethylenediamine modified chitosan	19.4	[62]
	PVA-CS/CE composite hydrogel beads	26.74	This study
**Co(II)**	Chitosan/poly(vinyl alcohol) magnetic composite	14.39	[67]
	Potassium hydroxide-modified activated carbon fiber	14.80	[68]
	Poly(vinyl alcohol)/chitosan nanofiber membrane	17.80	[69]
	PVA-CS/CE composite hydrogel beads	20.53	This study

## Data Availability

Not applicable.

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
