# Peer review of "Synthesis and Characterization of Biodegradable Poly(vinyl alcohol)-Chitosan/Cellulose Hydrogel Beads for Efficient Removal of Pb(II), Cd(II), Zn(II), and Co(II) from Water"

_gels, 2023, doi:10.3390/gels9040328_

Round 1
Reviewer 1 Report
Please find my comments in the attached file.

Reviewer 3 Report
The manuscript has good results and is attractive for the readers of this journal. However, it requires some critical revisions before possible publication in the journal of Gels. My important comments for promoting the quality of the manuscript are given below:
1- The novelty should be briefly presented in the abstract. Also, its innovation should be presented at the end of the introduction.
2- What is important factors on the sorption process? These critical factors should discuss in the introduction.
3- Sections 2.3.1., 2.3.2., and 2.3.3: It does not need to explain each analysis in separate section. All analyses should be explained in one paragraph. Revise it.
4- Section 2.3.4 should be explained more.
5- More details should be added to section 2.4. What is the studied factors? What about the analysis method of metal ions? And other important details need to be explained.
6- Have you used deionized water as leaching agent to remove heavy metals from the sorbent surface? Please explain. Why have methanol or acetone not used for this purpose?
7- Section 2.7. Adsorption kinetics: The equations in this section has no reference. Use the following paper for citation of this section: Bai, B., Bai, F., & Sun, S. Adsorption mechanism of shell powders on heavy metal ions Pb2+/Cd2+ and the purification efficiency for contaminated soils. Frontiers in Earth Science (2023).
Also, the following article can be used for isotherm interpretation: Radionuclide transport in multi-scale fractured rocks: A review. Journal of hazardous materials.
8- XRD and FTIR analyses need to be better explained and cited. Chen, Y., Li, J., Lu, J., Ding, M., & Chen, Y. (2022). Synthesis and properties of Poly(vinyl alcohol) hydrogels with high strength and toughness. Polymer Testing; Simultaneous removal of sulfate and nitrate from real high-salt flue gas wastewater concentrate via a waste heat crystallization route. Journal of Cleaner Production; Catalytic ozonation mechanisms of Norfloxacin using Cu–CuFe2O4. Environmental Research.
9- What is the particle size of the adsorbent? Can you calculate from SEM analysis?
10- What is the main morphology of the adsorbent? As seen from Fig. 4, similar morphology in all images? Also, the active sites on the surface of the adsorbent is not seen.
11- Alongside SEM analysis, the authors have presented EDX analysis for the adsorbent before and after adsorption. However, in the title of Fig. 4, EDX analysis has not been mentioned. Please revise the text.
12- For interpretation of the results, the authors can read and use the following papers: Innovative and green utilization of zinc-bearing dust by hydrogen reduction: Recovery of zinc and lead, and synergetic preparation of Fe/C micro-electrolysis materials. Chemical Engineering Journal; The remediation efficiency of heavy metal pollutants in water by industrial red mud particle waste. Environmental Technology & Innovation; Novel method for high-performance simultaneous removal of NOx and SO2 by coupling yellow phosphorus emulsion with red mud. Chemical engineering journal; Ultrasonic power combined with seed materials for recovery of phosphorus from swine wastewater via struvite crystallization process. Journal of environmental management.
13- The reusability of this study should be compared with similar studies. At least 2 works.
14- The quality and font size should be improved in Fig. 8.
15- Compare the maximum sorption capacity of this adsorbent with other adsorbents. I think this factor is not high compare to other adsorbents. Do you think your work is economical? Do you calculate the economic viability of this study?
16- Future important perspectives should be provided in the conclusion.
Reviewer 4 Report
The paper constitutes a really good, well-prepared and well-organized work. The whole has been presented very well. The conclusions drawn based on the performed investigations are promising. The only minor revisions which are suggested are as follows:
1) The notation of polymer names should be corrected, e.g. in the title of the paper it should be "poly(vinyl alcohol)" instead of "polyvinyl alcohol".
2) Section 3.1.: Authors mentioned: "Moreover, the incorporation of CE to a CS structure produced denser and mechanically stronger hydrogel beads" - such a conclusion was drawn only based on the visual observation? If yes, maybe it is worth to consider mechanical studies, e.g. determining the hardness of obtained beads via Shore hardness test.
3) Figure 3a) - from the editorial viewpoint, the text overlaps the figure - it needs to be corrected.
4) Section References should be corrected and prepared in line with the requirements of the Journal. For example, currently some references contain the whole journal names, and the other ones contain their abbreviations - it should be consistent.
Reviewer 5 Report
The work presents a well-conducted and thorough study on the characterizations of PVC-CS/CE hydrogels. The manuscript was well-written and constructed. The reviewer recommends the work to publish with the requests of additional discussions.
1. As mentioned in title and other sections, the hydrogels are biodegradable. The reviewer suggests adding some discussion regarding the biodegradation processing.
2. The reviewer can understand the hydrogels can be biodegradable. However, once the hydrogels absorbed the heavy metals, the heavy metals cannot be biodegradable. Please add some discussion regarding to this point.
3. Minor comment: Figure 4 EDX spectrums have inserted tables which can not be clearly read. The reviewer suggests to separate the table out from the figure.
4. The subfigures in Figure 7 have different font sizes and distortion. If possible, please adjust the fonts to be consistent.
5. Some final discussions are needed to add regarding the broad impact of the study and the potential future topic based on the work.
Round 2
Reviewer 3 Report
The authors have made all corrections properly and the manuscript has been modified carefully. Therefore, it can be published in Gels.